# Fast Detection of Two Smenamide Family Members Using Molecular Networking

**DOI:** 10.3390/md17110618

**Published:** 2019-10-30

**Authors:** Alessia Caso, Germana Esposito, Gerardo Della Sala, Joseph R. Pawlik, Roberta Teta, Alfonso Mangoni, Valeria Costantino

**Affiliations:** 1Dipartimento di Farmacia, Università degli Studi di Napoli Federico II, via Domenico Montesano 49, 80131 Napoli, Italy; alessia.caso@unina.it (A.C.); germana.esposito@unina.it (G.E.); roberta.teta@unina.it (R.T.); alfonso.mangoni@unina.it (A.M.); 2Laboratory of Pre-clinical and Translational Research, IRCCS-CROB, Referral Cancer Center of Basilicata, Rionero in 85028 Vulture, Italy; gerardo.dellasala@unina.it; 3Department of Biology and Marine Biology, University of North Carolina Wilmington, Center for Marine Science, 5600 Marvin K Moss Lane, Wilmington, NC 28409, USA; pawlikj@uncw.edu

**Keywords:** *Smenospongia aurea*, marine natural products, structure elucidation, anti-tumor lead molecules, smenamides, solid tumor cell lines, conformational analysis

## Abstract

Caribbean sponges of the genus *Smenospongia* are a prolific source of chlorinated secondary metabolites. The use of molecular networking as a powerful dereplication tool revealed in the metabolome of *S. aurea* two new members of the smenamide family, namely smenamide F (**1**) and G (**2**). The structure of smenamide F (**1**) and G (**2**) was determined by spectroscopic analysis (NMR, MS, ECD). The relative and the absolute configuration at C-13, C-15, and C-16 was determined on the basis of the conformational rigidity of a 1,3-disubstituted alkyl chain system (i.e., the C-12/C-18 segment of compound (**1**). Smenamide F (**1**) and G (**2**) were shown to exert a selective moderate antiproliferative activity against cancer cell lines MCF-7 and MDA-MB-231, while being inactive against MG-63.

## 1. Introduction

Marine organisms are a source of compounds with enormous chemical diversity, which, in turn, translates to a wide variety of biological activities. Many marine natural products show unusual and unique chemical structures, often containing halogen atoms. Halogenated natural products are widely present in nature. Currently, more than 5000 compounds are reported in the literature and most of them have a marine origin [1,2]. 

Our research program, aimed at discovering new bioactive compounds from marine organisms, focused in recent years on the metabolome of Caribbean sponges of the genus *Smenospongia*, which have proven to be very rich in new chlorinated secondary metabolites. Among them are smenamides [3], smenothiazoles [4] and conulothiazoles [5], belonging to the hybrid peptide/polyketide chlorinated class of compounds; smenolactones, four chlorinated compounds with a polyketide structure [6]; and smenopyrone, a biogenetically different compound with a polypropionate structure containing two γ-pyrone rings [7].

Smenamide A (**3**) (Figure 1) isolated in 2013 from the Caribbean sponge *Smenospongia aurea*, showed potent cytotoxic activity at nanomolar levels on lung cancer Calu-1 cells through an unambiguous pro-apoptotic mechanism [3].

Although the cytotoxic activity of smenamide A could be determined with only a few micrograms of the compound, larger amounts were needed to further study the biological activity and to determine the configuration at C-16, which remained undetermined until the total synthesis of 16-*epi*-smenamide A and *ent*-smenamide A was accomplished [8]. To further our understanding of the activity-related structural features of this class of molecules, a series of short derivatives of the 16-*epi*- series were designed, prepared, and tested for antiproliferative activity [9]. Recently, the family of smenamide compounds was further expanded with the isolation of smenamides C, D, and E. Interestingly, smenamides C (**5**) and E (**7**) showed moderate neurotoxicity against neuro-2A cells, while smenamide D (**6**), the geometric isomer of **5**, did not show any cytotoxic activity [10]. Moreover, it is interesting to cite a recent publication by Cantrell et al. in which molecular networking in combination with MS2LDA analysis allowed to describe the intra and inter-chemical diversity present in a *S. aurea* sample collected in Florida Keys [11].

In this paper, we report on the use of a molecular networking dereplication strategy which resulted in the rapid detection from extracts of *S. aurea* of two new members of the smenamide family, smenamide F (**1**) and G (**2**). The new compounds were isolated, their structure elucidated, and their absolute configuration determined. Smenamide F (**1**) was shown to be a hydrated analog of smenamide A (**3**). Smenamide G (**2**) only differs from (**1**) in the configuration at C-8 (the *α* carbon of the putative biosynthetic precursor phenylalanine). Despite the minute amounts isolated, a preliminary biological screening was performed on a panel of solid tumour cell lines [MCF-7 (breast cancer), MDA-MB-231 (triple negative breast cancer), and MG-63 (osteosarcoma)] to investigate the possible antiproliferative activity of the two compounds.

## 2. Results and Discussion

### 2.1. Fast Detection of Two New Analogues of the Smenamide Family: Construction of the Molecular Network and Isolation

A specimen of *S. aurea* collected along the coast of Little Inagua (Bahamas Islands) was extracted as described in our previous papers [3,4,5,12]. Briefly, the sponge was extracted in sequence with MeOH, MeOH/CHCl_3_ mixtures, and CHCl_3_. The methanol extract was partitioned between H_2_O and BuOH, and the BuOH layer was combined with the other organic extracts. The crude extract was subjected to C-18 reversed-phase column chromatography. The fraction containing chlorinated metabolites was partitioned in a two-phase system composed of chloroform, methanol and acidic water (1% *v/v* of acetic acid) to remove the high amounts of brominated alkaloids also present in the sponge. The organic phase was analyzed for chlorinated metabolites by LC-HRMS and LC-HRMS/MS on a LTQ Orbitrap instrument, and the raw analytical data were elaborated and studied using molecular networking. Molecular networking is a powerful bioinformatic tool used to speed up dereplication of compounds from the organic extract, leading to the identification of known metabolites as well as new analogues [13,14]. Specifically, raw LC-MS data were first pre-processed using the mzMine program [15] to deconvolute chromatographic peaks, distinguish between isomeric compounds based on their retention times, and remove isotope and adduct peaks, as described in details in our previous paper [6]. Then, the .mgf MS2 data files were submitted to the Global Natural Product Social Molecular Network (GNPS) online platform [16], along with the reference MS2 spectra of smenamides C and E [10], and a molecular network was generated. Finally, the network was visualized using the Cytoscape program [17]. Two smenamide-related clusters were present in the network and are shown in Figure 2. Cluster 1 contained smenamide A and B, along with smenamide C and two minor unidentified isomeric compounds at *m/z* 487.23 previously detected and described in reference [11]. Cluster 2 contained smenamide E and three more compounds, smenamides F (**1**) and G (**2**), and an unidentified analogue at *m/z* 533.28. These nodes were also present in the network reported in [11], although they are not discussed in the text.

Interestingly, although compounds 1–7 all belong to the smenamide family, they are split into two different clusters, according to the hydration state of the central part of the molecule. Therefore, it is evident that the hydration of the double bond in position 13 affects in a very significant way the fragmentation of smenamide. A cluster containing the new smenamides F and G was also present in the network comparing chlorinated metabolites of *S. aurea* and the *cyanobacterium Trichodesmium* sp. [6]; the network showed they were present exclusively (smenamide G) or almost exclusively (smenamide F) in *S. aurea*.

The new smenamides F and G were isolated by subjecting the chlorinated metabolite fraction to two subsequent reversed-phase HPLC separations, which afforded smenamide F (**1**, 237 μg) and G (**2**, 91 μg) as pure compounds.

### 2.2. Structure Elucidation of Smenamides F and G

The high-resolution ESI MS spectrum of smenamide F (**1**) displayed peaks at *m/z* 519.2620 ([M + H]^+^) and 541.2440 ([M + Na]^+^). Intense (32%) M+2 isotope peaks of both ions suggested the presence of one chlorine atom in the molecule, allowing for the determination of the molecular formula C_28_H_39_ClN_2_O_5_ with 10 unsaturations. The ^1^H NMR spectrum of smenamide F (**1**) was indicative of a mixed NRPS/PKS biogenesis metabolite, in that it showed signals for five aromatic protons, suggesting a monosubstituted phenyl group, one amino acid *α* proton (H-8, *δ* 4.90), two olefinic protons (H-10, *δ* 4.90 and H-21, *δ* 5.97), five methyl groups (one acetyl methyl, one *N*-methyl, one *O*-methyl and two aliphatic methyl groups), and a methine proton (H-15, *δ* 3.99). The general features of the spectrum suggested compound 1 to belong to the smenamide family. A second methyl doublet (H_3_-14, *δ* 1.02) replacing the vinylic methyl singlet of smenamide A (**3**), the oxymethine group at C-15, and one less unsaturation showed smenamide F (**1**) to be a hydrated analog of smenamide A (**3**). 

In the polyketide moiety of the molecule, a moderately deshielded methylene group (H_2_-22, *δ* 2.25) showed a COSY correlation peak with a second methylene group (H_2_-23, *δ* 1.71) which was, in turn, *J*-coupled with another methylene group (H_2_-24, *δ* 3.39). This latter group showed HMBC correlation peaks with an *N*-methyl group (CH_3_-27, *δ_H_* 3.07, *δ_C_* 36.7) and a carbonyl carbon (C-25, *δ_C_* 173.4). The presence of correlation peaks of the *N*-methyl group at C-27, and an acetyl methyl group (CH_3_-26, *δ_H_* 2.09, *δ_C_* 21.8) with the carbonyl carbon atom at *δ_C_* 173.4, established the *N*-methylacetamido function as the western terminus of the molecule. As a result of the *E/Z* conformational equilibrium at the *N*-methylacetamido function, most resonances in the proton NMR spectrum of compound **1** were split into two signals, with an approximate 1:1 ratio. This is quite common for cyanobacterial metabolites with a tertiary *N*-methyl amide function, such as kalkitoxin [18]. The Z conformer was identified from a strong ROESY correlation peak between H_3_-26 (*δ* 2.09) and H_3_-27 (*δ* 3.07), whereas the E conformer showed a correlation peak between H_3_-26 (*δ* 2.11) and H_2_-24 (*δ* 3.39) (Figure 3). Another deshielded methylene function (H_2_-19, *δ* 2.21) was correlated by COSY to a diasterotopic pair of methylene protons (H_a_-18, *δ* 1.64; H_b_-18, *δ* 1.45), in turn coupled with the methine H-16 (*δ* 1.66). The latter showed COSY correlation peaks with a methyl group (H_3_-17, *δ* 0.94) and with the H-15 oxymethine proton at *δ* 3.99. Finally, H-15 was coupled with the methine proton H-13 at *δ* 4.08, in turn coupled with another methyl group (H_3_-14, *δ* 1.02). The HMBC correlation peaks of H_2_-19 and H_2_-22 with the methine carbon at *δ* 113.8 and the non-protonated carbon at *δ* 143.7 identified a trisubstituted double bond, linking the two partial structures defined up to this point. The deshielded chemical shift of H-21, along with the shielded chemical shift of C-21 (*δ* 113.8), revealed the presence of a chlorine atom linked to C-21, in agreement with the molecular formula. The remaining part of the molecule was easily recognized as a dolapyrrolinone unit based on the similarity of ^1^H and ^13^C NMR profiles of smenanamide F (**1**) with smenamide A (**3**), along with the overall HMBC spectral data (Figure 3).

Smenamide F (**1**) contains one double bond at C-20/C-21 and four stereogenic centers at C-8 (the amino acidic *α* carbon), C-13, C-15, and C-16. The Z configuration at position 20 was determined from the ROESY correlation peak between H-21 and H_2_-19 (*δ* 2.21). The configuration at C-8 was determined by Marfey’s method [19]. Compound **1** was subjected first to ozolysis to prevent racemization at the *α* carbon of the pyrrolidinone-modified amino acid [20], then to hydrolysis with HCl 6 N, and finally to derivatization with the l enantiomer of Marfey’s reagent [*N*-(5-fluoro-2,4-dinitrophenyl)-l-alaninamide, or l-FDAA]. The resulting l-FDAA derivative was subjected to LC-ESIMS analysis and showed the same retention time of an authentic standard prepared from l-FDAA and l-Phe, thus determining the S configuration at C-8.

The conformational rigidity of a 1,3-disubstituted alkyl chain (like the C-12/C-18 segment of compound 1) was the foundation for determining the relative configuration at C-13, C-15, and C-16. As discussed in our previous work [6], for such a system, only two low-energy conformers may exist (Figure 4A), because of the unfavorable syn-pentane interactions [21] present in all the other conformers. In addition, for this system, the expected pattern of couplings between vicinal protons is opposite between the two conformers (Figure 4A). Therefore, the large coupling between H-13 and H-15 and the small coupling between H-15 and H-16 showed that one of the two conformers must be largely predominant in compound 1 (irrespective of configurations at C-13, C-15, and C-16). 

Under these assumptions, the relative configurations at C-13 and C-16 could be determined based on NOESY data. The two possible diastereomers are depicted in Figure 4B,C. Both conformations are represented for each diastereomer, while configuration at C-15 is left unassigned for the time being. It can be easily seen that experimental data, i.e., the strong NOESY correlations of H-13 with H_3_-17 and H-16 with H_3_-14 and the absence of any correlation of H-13 with H_2_-18, could only be accounted for by conformation d in Figure 4C. This established both the C13/C16 relative configuration and the conformation of this part of the molecule. On this rigid framework, configuration at C-15 could be easily determined on the basis of vicinal couplings as shown in Figure 4D. This was confirmed by the NOESY correlation between the gauche vicinal protons H-15 and H-16, and by the absence of NOE between the anti-vicinal protons H-13 and H-15. Therefore, the configuration of the polyketide part of the molecule was assigned as 13*S*,15*S*,16*R* or 13*R*,15*R*,16*S*. 

Unfortunately, we were not able to correlate configuration at C-8 with configurations at C13/C15/C/16. However, with the reasonable hypothesis that smenamide F (**1**) possesses the same 16R configuration as smenamide A (**3**) [8], the absolute stereochemistry of compound **1** can be assumed to be 8*S*,13*S*,15*S*,16*R*,20*Z*. 

The high-resolution ESI mass spectrum of smenamide G (**2**) showed [M + H]^+^ and [M + Na]^+^ ions at *m*/*z* 519.2620 and 541.2440, respectively, consistent with the molecular formula C_28_H_39_ClN_2_O_5_, and suggesting that it is isomeric with smenamide F (**1**). The one-dimensional ^1^H NMR spectrum of compound **2** appeared nearly identical to that of compound **1** (including conformational equilibrium), with minor differences in chemical shifts for all signals. Correlation peaks in the 2D NMR spectra (including NOESY) were also very similar, and the MS/MS fragmentation pattern was identical (Appendix A). These observations led to the conclusion that smenamide F and G were stereoisomers. The coupling constants of protons at C-13, C-15, and C-16 were very similar for both compounds, suggesting that smenamide G (**2**) has the same relative configuration as smenamide F (**1**). In contrast, Marfey’s analysis showed that the l-FDAA derivative obtained from compound **2** was identical to l-FDAA-d-Phe, and therefore, that configuration at C-8 was *R*. Therefore, smenamide G (**2**) was determined as the epimer at C-8 of smenamide F (**1**), i.e., the 8*R*,13*S,*15*S,*16*R*,20*Z* stereoisomer.

To support the stereochemical assignments made so far, the ECD spectra of smenamides F (**1**) and G (**2**) were measured. The ECD spectrum of smenamide F (Appendix A) was similar to that of smenamide A (**3**), while the ECD spectrum of smenamide G (Appendix A) was approximately its mirror image. This data clearly showed that the ECD spectrum of the smenamides is dominated by the chirality of the dolapyrrolinone moiety, and therefore, cannot provide information on the configuration of the polyketide part of the molecule.

### 2.3. Antiproliferative Activity of Smenamides F and G

Preliminary screening of antiproliferative activity of smenamide F and G was performed on three different solid tumor lines, namely MCF-7 (breast cancer), MDA-MB-231 (triple negative breast cancer) and MG-63 (osteosarcoma) cell lines. Smenamides F and G were evaluated individually against cancer cells at concentrations of 1 and 5 μM for 96 h. As previously reported [6], cell growth was monitored in real-time by using the xCELLigence System Real-Time Cell Analyzer (RTCA), which translates electronic impedance variations into cell index (CI) data, a closely related parameter to cell viability and morphology. At the lowest dose (1 μM) of each compound, cell proliferation was unaffected or only slightly delayed after drug treatment in all three models. At 5 μM concentration, smenamide F (**1**) and G (**2**) were shown to exert selective moderate antiproliferative activity against both MCF-7 and MDA-MB-231 breast carcinoma cell lines (Figure 5), while being inactive against osteosarcoma cells (Figure 6). Indeed, smenamides prompted a) significant reduction in the slope of the growth curve (within the range of 30–40%) and b) significant delay in cell doubling time exclusively in breast models at the highest dose (5 μM). Moreover, the absolute configuration at C-8 of the dolapyrrolinone unit does not affect growth inhibitory properties of compounds in the tested cell lines.

## 3. Materials and Methods 

### 3.1. General Experimental Procedures

High-resolution ESI-MS and HR-ESI-MS-HPLC experiments were performed on a Thermo LTQ Orbitrap XL mass spectrometer (Thermo Fisher Scientific Spa, Rodano, Italy) coupled to an Agilent model 1100 LC system (Agilent Technology Cernusco sul Naviglio, Italia). The spectra were recorded by infusion into the ESI source using MeOH as the solvent. CD spectra were recorded using a Jasco J-710 (Easton, MD) spectrophotometer using a 1 mm cell. NMR experiments were performed on Varian Unity Inova spectrometers (Agilent Technology Cernusco sul Naviglio, Italia) at 700 MHz in CD_3_OD; chemical shifts were referenced to the residual solvent signal (CD_3_OD: *δ_H_* 3.31, *δ_C_* 49.00). All ^13^C chemical shift were assigned using the 2D spectra, therefore, mono-dimensional ^13^C NMR spectra were not recorded (see Appendix A). For an accurate measurement of the coupling constants, the one-dimensional ^1^H NMR spectra were transformed at 64 K points (digital resolution: 0.09 Hz). Through-space ^1^H connectivities were evidenced using a ROESY experiment with a mixing time of 450 ms. The HSQC spectra were optimized for *^1^J_CH_* = 142 Hz and the HMBC experiments for *^2,3^J_CH_* = 8.3 Hz. High-performance liquid chromatography (HPLC) separations were achieved on a Varian Prostar 210 apparatus equipped with a Varian Prostar 325 UV-Vis detector. 

### 3.2. Collection, Extraction and Isolation

A specimen of *Smenospongia aurea* was collected by scuba along the southwest coast of Little Inagua (Bahamas Islands) on 9 July 2013 on a research expedition using the *R/V Walton Smith*. After collection, the sample was identified onboard following the information reported on the website The Sponge Guide [22], immediately frozen and stored at −20 °C until extraction. The sponge (712 g wet weight) was homogenized and extracted with MeOH (4 × 4 L), MeOH and CHCl_3_ in different ratios (2:1, 1:1, 1:2) and then with CHCl_3_ (2 × 4 L). The MeOH extract was partitioned between H_2_O and *n*-BuOH; the BuOH layer was combined with the CHCl_3_ extracts and concentrated in vacuo.

The resulting organic extract (16.31 g) was subjected to reversed-phase chromatography using a column packed with RP-18 silica gel. The fraction eluted with MeOH/H_2_O (9:1) (363.7 mg) was partitioned in a two-phase system composed by H_2_O (160 mL), MeOH (260 mL), CHCl_3_ (140 mL) and AcOH (5 mL). The organic layer, containing smenamides, was subjected to reversed-phase HPLC separation (column 250 × 10 mm, 10 μm, Luna (Phenomenex) C18; Eluent A: H2O; Eluent B: MeOH; gradient: 55% → 100% B over 60 min, flow rate 5 mL/min), thus affording a fraction (*tR* = 27.5 min) containing Compound 1 and a fraction (*tR* = 28.5 min) containing Compound 2. The two fractions were each separated on reversed-phase HPLC (column 250 × 4.6 mm, 5 μm, Luna (Phenomenex) C18; Eluent A: H2O; Eluent B: ACN; gradient: 50%→100% B, over 35 min, flow rate 1 mL∙min^−1^), which gave 237 µg of pure compound **1** (*tR* =14.0 min) and 91 μg of pure compound **2** (*tR* = 14.5 min).

#### 3.2.1. Smenamide F (**1**)

Colorless amorphous solid, HRESIMS (positive ion mode, MeOH) *m/z* 541.2443 ([M + Na]^+^, C_28_H_39_ClN_2_O_5_ calcd. 541.2445); MS isotope pattern: M (100%), M + 1 (30%, calcd. 30.9%), M + 2 (34%, calcd. 32%); HRESIMS/MS (parent ion *m/z* 541.2443, C_28_H_39_ClN_2_NaO_5_^+^): *m/z* 523.2332 (C_28_H_37_ClN_2_NaO_4_^+^, calcd. 523.2334), *m/z* 294.1600 (C_15_H_26_ClNNaO^+^, calcd. 294.1595), *m/z* 282.1252 (C_13_H_22_ClNNaO_2_^+^, calcd. 282.1231). 

^1^H and ^13^C NMR: Appendix A. CD (MeOH): *λmax* 229 (+18.3). 

#### 3.2.2. Smenamide G (**2**)

Colorless amorphous solid, HRESIMS (positive ion mode, MeOH) *m/z* 541.2440 ([M + Na]^+^, C_28_H_39_ClN_2_O_5_ calcd. 541.2445); MS isotope pattern: M (100%), M + 1 (30%, calcd. 30.9%), M + 2 (34%, calcd. 32%); HRESIMS/MS (parent ion *m/z* 541.2443, C_28_H_39_ClN_2_NaO_5_^+^): *m/z* 523.2338 (C_28_H_37_ClN_2_NaO_4_^+^, calcd. 523.2334), *m/z* 294.1610 (C_15_H_26_ClNNaO^+^, calcd. 294.1595), *m/z* 282.1249 (C_13_H_22_ClNNaO_2_^+^, calcd. 282.1231). 

^1^H and ^13^C NMR: Appendix A. CD (MeOH): *λmax* 228 (−6.9).

### 3.3. Determination of the Absolute Configuration of the Amino Acids

#### 3.3.1. Ozonolysis and Hydrolysis

A small amount of compound **1** (5 μg) and **2** (5 μg) were separately suspended in ozone-saturated MeOH (300 μL) at −78 °C for 5 min. The samples were dried under a N_2_ stream to remove the ozone, then treated with 6 N HCl and heated in a flame-sealed glass tube at 180 °C for 2 h. The residual HCl fumes were removed in vacuo.

#### 3.3.2. Marfey’s Derivatization with D- and L-FDAA

The hydrolysate of **1** was dissolved in triethylamine/ACN (2:3) (80 µL), and this solution was then treated with 1% 1-fluoro-2,4-dinitrophenyl-5-l-alaninamide (l-FDAA) in ACN/acetone (1:2) (75 μL). The vial was heated at 50 °C for 1 h. The mixture was dried and resuspended in ACN/H_2_O (5:95) (500 μL) for subsequent LC-MS analysis (I). The hydrolysate of **2** was treated with d-FDAA under identical conditions as **1** (II). Authentic l-Phe standard was treated with l-FDAA and d-FDAA as above (III).

#### 3.3.3. LC-MS Analysis

The Marfey’s derivatives were analyzed by HR-ESI-MS-HPLC. A 5 μm Kinetex C18 column (100 × 2.10 mm), maintained at room temperature, was eluted at 250 μL·min^−1^ with H_2_O and ACN, using a gradient elution. The gradient program was the following: 5% ACN 3 min, 5–60% ACN over 20 min, 90% ACN 5 min. Mass spectra were acquired in positive ion detection mode, and the data were analyzed using the suite of programs, Xcalibur. The retention times of the Marfey’s derivatives (I), (II) and (III) were the following, *t_R_* = (l/d) in min: phenilalanine (*m/z* 418.1357, [M + H]^+^) *t_R_* = 17.32/18.60; l-FDAA-Phe from **1**
*t_R_* = 17.68; l-FDAA-Phe from **2**
*t_R_* = 18.63.

### 3.4. Molecular Network

A molecular network [14] was generated using the online workflow at GNPS (https://gnps.ucsd.edu/). The parent mass tolerance and MS/MS fragment ion tolerance were both set at 0.02 Da. In the resulting molecular network, edges were refined to have a cosine score above 0.6 and more than six matched peaks. The spectra in the network were then compared with those in GNPS spectral libraries. To observe a matching between network spectra and library spectra, a cosine score above 0.7 and at least six matched peaks were required. Once the basic molecular network was generated, it was visualized with Cytoscape 3.2.122 [15].

### 3.5. Cell Culture

MCF-7, MDA-MB-231 and MG-63 cells were purchased from American Type Culture Collection (ATCC, Manassas, VA, USA). MCF-7, MDA-MB-231 and MG-63 cells were cultured in DMEM medium at 37 °C in a 5% CO_2_ humidified atmosphere. DMEM medium was supplemented with 10% fetal bovine serum, penicillin–streptomycin (100 U/mL), and 2 mM *L*-glutamine. Cell morphology was observed using an inverted optical microscope (Axio Vert A1, Zeiss, Oberkochen, Germany). When cells reached the confluence, cells were detached with 0.05% trypsin-EDTA to perform xCELLIgence assays.

### 3.6. xCELLigence Assays and Statistical Analysis

The xCELLigence System Real-Time Cell Analyzer (ACEA Biosciences, San Diego, CA, USA) was used for monitoring proliferation of cancer cells. Antiproliferative assays were performed as previously described [6]. MCF7 and MDA-MB-231 cells were seeded at a cell density of 3000 cells/well while MG-63 cells were seeded at a cell density of 4000 cells/well. Approximately 24 h after seeding, medium was removed, and cells were treated with medium containing either 1 or 5 μM concentrations of each compound for 96 h. 

For our data analysis, CI was normalized just before drug treatment and converted into a normalized cell index (NCI). Normalized cell index was calculated as follows: NCI = CI _end of treatment_/CI _normalization time_. Real-time NCI traces were produced by the RTCA-integrated software. Antiproliferative effects of smenamides are presented either as cell index slopes relative to controls treated with DMSO vehicle or as cell doubling times. Cell index slopes and doubling times were calculated using the RTCA-integrated software. The NCI slope describes the rate of change of the cell index for cells after treatment with a cytotoxic compound. Doubling time is the time required for a curve cell index value to double. Data are presented as mean ± standard deviation (*n* = 3). Differences between groups were determined by analysis of variance (ANOVA) to determine statistical significance, which is reported as *p* value. The Dunnett’s Multiple Comparison Test was used to compare treatments with controls. Statistical analysis was performed using the GraphPad Prism Software Version 5.

## 4. Conclusions

The use of molecular networking as a dereplication strategy allowed for the rapid detection of two new members of the smenamide family of compounds, smenamide F (**1**) and G (**2**). A preliminary biological screening was performed on three solid tumor cell lines [MCF-7 (breast cancer), MDA-MB-231 (triple negative breast cancer) and MG-63 (osteosarcoma)] to evaluate the possible antiproliferative activity of the compounds. Data obtained showed that smenamide F (**1**) and G (**2**) exert a selective moderate antiproliferative activity against both breast carcinoma cell lines (MCF-7 and MDA-MB-231) at 5 μM, while being inactive against the MG-63 osteosarcoma cell line.

## Figures and Tables

**Figure 1 marinedrugs-17-00618-f001:**
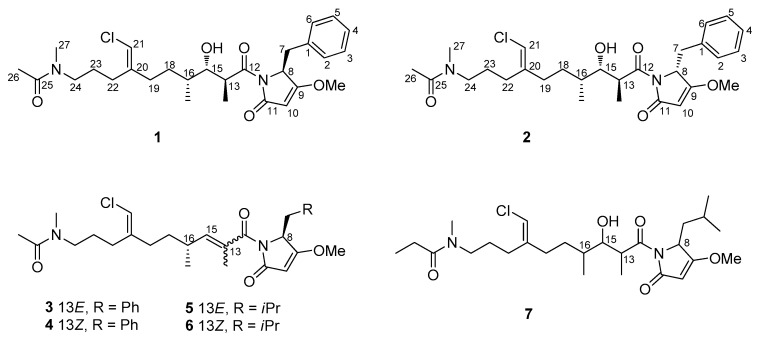
Structures of smenamide F (**1**), smenamide G (**2**), and smenamides A–E (**3**–**7**).

**Figure 2 marinedrugs-17-00618-f002:**
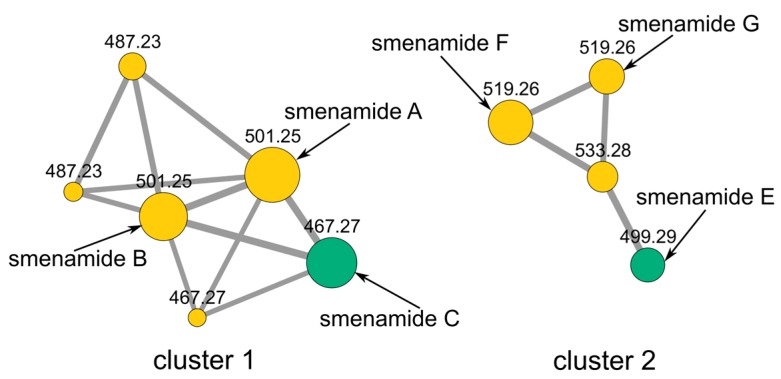
The molecular network obtained from the organic extract of *Smenospongia aurea*. Nodes are labeled with parent *m/z* values. Node size is indicative of the ion count; edge thickness is relative to cosine score.

**Figure 3 marinedrugs-17-00618-f003:**
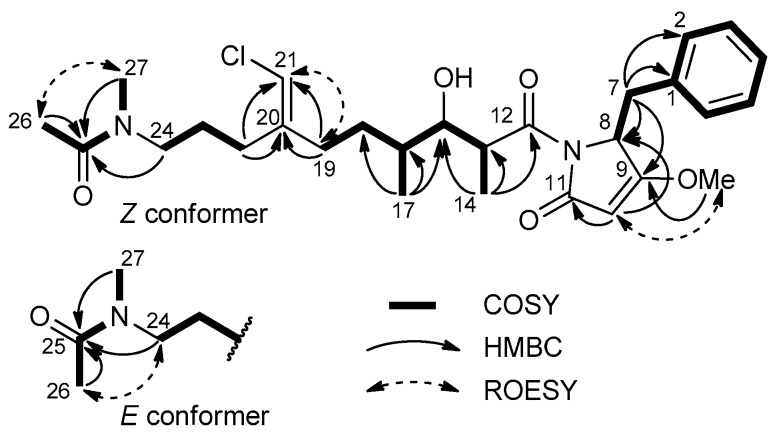
Most significant correlations provided by the COSY, HMBC, and ROESY 2D NMR spectra of smenamide F (**1**).

**Figure 4 marinedrugs-17-00618-f004:**
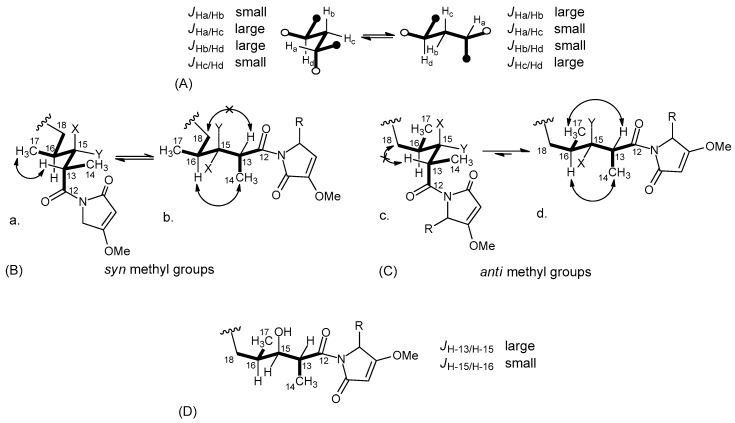
(**A**) The only two possible conformations of a 1,3-disubstituted alkyl chain; (**B**) Conformation with syn methyl groups; (**C**) Conformation with anti-methyl groups; (**D**) Configuration of the C-12/C-18 polyketide portion of the molecule.

**Figure 5 marinedrugs-17-00618-f005:**
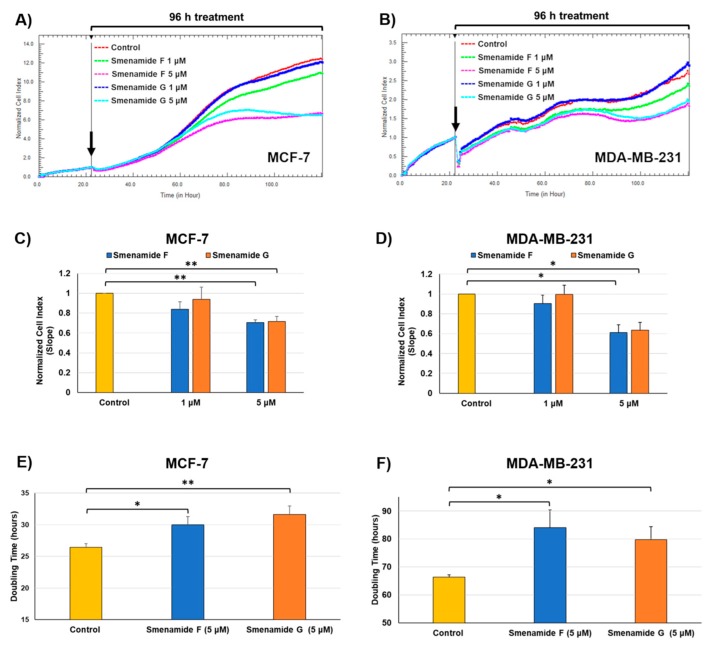
Real-time dynamic monitoring of breast cancer cells proliferation after treatment with smenamides F and G, using the RTCA platform. (**A**,**B**) Normalized cell index (NCI) kinetics of MCF-7 (**A**) and MDA-MB-231 (**B**) cells after exposure to different concentrations (1 μM and 5 μM) of smenamides F and G, and to DMSO vehicle control for 96 h. Black arrows show the starting point of drug treatment. Each cell index value was normalized just before treatment. (**C**,**D**) NCI variations of MCF-7 (**A**) and MDA-MB-231 (**B**) cells after 96 h exposure to different concentrations (1 and 5 μM) of smenamides F and G, and to DMSO vehicle control. Antiproliferative effects are reported as the slope of NCI to describe the changing rate of growth curves after drug treatment. NCI slope values are relative to controls treated with DMSO vehicle. (**E**,**F**) Doubling times of NCI of MCF-7 (**E**) and MDA-MB-231 (**F**) cells after exposure to 5 μM of smenamides F and G, and 0.5% DMSO. Data are presented as mean ± SD; *n* = 3. * *p* < 0.05; ** *p* < 0.01.

**Figure 6 marinedrugs-17-00618-f006:**
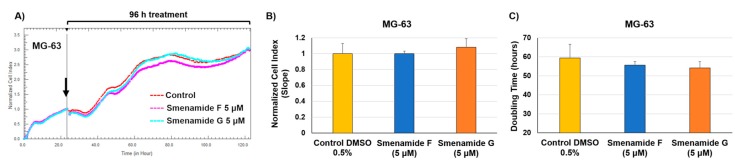
Real-time dynamic monitoring of osteosarcoma cancer cells (MG-63) proliferation after treatment with smenamides F and G, using the RTCA platform. (**A**) Normalized cell index (NCI) kinetics of MG-63 cells after exposure to 5 μM of smenamides F and G, and to 0.5% DMSO vehicle control for 96 h. Black arrow shows the starting point of drug treatment. Each cell index value was normalized just before treatment. (**B**) NCI variations of MG-63 cells after 96 h exposure to 5 μM of smenamides F and G, and to 0.5% DMSO vehicle control. Antiproliferative effects are reported as the slope of NCI to describe the changing rate of growth curves after drug treatment. NCI slope values are relative to controls treated with DMSO vehicle. (**C**) Doubling times of NCI of MG-63 after exposure to 5 μM of smenamides F and G, and 0.5% DMSO. Data are presented as mean ± SD; *n* = 3.

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
