# Peer review of "Fast Detection of Two Smenamide Family Members Using Molecular Networking"

_marinedrugs, 2019, doi:10.3390/md17110618_

Round 1
Reviewer 1 Report
The publication by Caso et al. describes two new analogs of previously known class of compounds “smenamides” and their antiproliferative activity. The reviewer comments for this submission are below:
Lines 50-51. The authors should cite a highly relevant study “Cantrell, T.P., Freeman, C.J., Paul, V.J. et al. J. Am. Soc. Mass Spectrom. (2019) 30: 1373. https://doi.org/10.1007/s13361-019-02207-5”, which also uses molecular networking to describe chemical diversity of smenamide compounds produced by the sponge Smenospongia aurea. The authors highlight the power of molecular networking but they can further utilize this power. For example, the authors should comment upon whether these compounds were detected in their recent study on Smenospongia aurea: Chem. Front., 2019,6, 1762-1774 (smenamide E cluster shown in Figure 1) and whether they are also present in Trichodesmium sp data in their Figure 1. The power of molecular networking also lies in the fact that large amounts of data generated by a laboratory or similar data generated by other researchers across the globe can be re-analyzed without much effort and with only a few clicks. Hence, it would be worthwhile to report whether these molecules are represented in the molecular networks generated by the authors themselves in the past. A step forward would be to describe the presence of these molecules in datasets specific to sponges available from public repositories such as MassIVE. The authors are already using the GNPS platform to generate networks and can import these public datasets to co-network public data with their own data with a few clicks (For this analysis, classical networking approach is utilized). If these molecules are not detected in public dataset of Smenospongia, the authors can highlight whether this is an effect of different extractions methods (authors did partition their extracts to remove brominated alkaloids, which may have enhanced the signal for the new analogs reported), differences in LC-MS methods or instruments or whether there may be variation in different populations of sponges collected from different spots or a combination of these factors. Lines 85-88 “Cluster 1, containing smenamide A and B, along with smenamide C and two minor unidentified compounds; cluster 2, containing smenamide E and three more compounds, two of which were identified as smenamides F and G.Among the minor components is a node with m/z 23, which has been putatively described in Cantrell, T.P., Freeman, C.J., Paul, V.J. et al. J. Am. Soc. Mass Spectrom. (2019) 30: 1373. The authors should compare their MS2 spectra with the one shown in article by Cantrell et all, cite this reference, and report that the node with m/z 487.23 is putatively described previously (putative since no NMR data was provided) such that this information can be propagated to readers. The isotope peaks in supplementary Figure S3 with m/z 520 and 521 are of same height, whereas in high resolution data, the peak with m/z 521 should be slightly taller. Just in case, authors are reporting isotopic pattern from MS2 data, they should report it from MS1 data. Otherwise, such a minor deviation from theoretical isotope pattern is acceptable for this publication, especially when they are supporting structures with NMR data. The authors should explain why in Figure S9, the isotopic distribution is incorrect. In the supplementary figures for MS/MS spectra, the authors should consider providing structures of observed MS2 peaks (wherever possible). Provided chemical structures of MS2 peaks adds value to the annotations and is the essence of information provided by MS/MS spectra. In the absence of C13 NMR data, the assignment of allylic carbons is contentious. The authors should state that C13-APT data is not available and whether 2D correlations are used for these annotations. Line 199: 1 and 5 uM. 1 appears to be bold, which if bold gives a wrong impression that the authors are pointing to compound 1. The authors should also consider depositing their mass spectrometry data to the MassIVE repository and MS/MS spectra of new analogues to the GNPS spectral library (not required, only a suggestion). Making data publicly available, especially post-publication, allows the GNPS community to derive larger scale correlations between metabolomes of different species, organisms, and environments and contribute to our holistic understanding of chemical diversity present in our marine environments. MarinLit currently has structures of only smenamide D, E, and C. The authors may consider contacting the MarinLit resource to explore if additional analogues described by them can be added (https://www.rsc.org/locations-contacts/contact-us/resources-tools/) and may consider to deposit new analogues in the dictionary of natural products repository as well.
Author Response
Reviewer 1
Line 50-51. We deeply apologize for the lacking of the reference "Cantrell T.P.; Freeman C.J.; Paul V.J., Agarwal V.; Garg N. Mass Spectrometry-Based Integration and Expansion of the Chemical Diversity Harbored Within a Marine Sponge. Am. Soc. Mass Spectrom. 2019, 30, 1373-1384. doi: 10.1007/s13361-019-02207-5", which describes the chemical diversity of the compounds produced by S. aurea. We add this study in the main text (line 53-55). Authors thanks for the suggestions and add on the text two new paragraph:Line 88-93 "Cluster 1 contained smenamide A and B, along with smenamide C and two minor unidentified isomeric compounds at m/z 487.23 previously detected and described in ref. 11. Cluster 2 contained smenamide E and three more compounds, smenamides F (1) and G (2), and an unidentified analogue at m/z 533.28. These nodes were also present in the network reported in ref. 11, although not discussed in the text"
Line 101-104 "A cluster containing the new smenamides F and G was also present in the network comparing chlorinated metabolites of S. aurea and the cyanobacterium Trichodesmium sp. [6]; the network showed they were present exclusively (smenamide G) or almost exclusively (smenamide F) in S. aurea."
All 13C chemical shift were assigned using the 2D spectra, therefore mono-dimensional 13C NMR spectra were not recorded (line 234). For clarity, tables n. 1 and 2 have been moved in the text.
In Figure S9, the isotopic distribution was not correct. In fact, the spectrum has been recorded, soon after the same sample was used for the NMR studies and some deuterium was still present. Authors performed a new mass experiment and in the updated version the new spectra are present in SI. In other words, figure S9 is new. Moreover, in Figure S2 and S9 the heights of the two isotopic peaks are the same. This can happen when using the LTQ/ Orbitrap mass spectrometer as reported in the following paper:
https://doi.org/10.1016/j.jasms.2009.07.014
Authors will deposit all mass data to the MassIve and GNPS, as suggested.
Reviewer 2 Report
The article entitled FAST DETECTION OF TWO SMENAMIDE FAMILY-MEMBERS USING MOLECULAR NETWORKING is well written and it describes exhaustively all the procedures to get the chemicals from the Sponge Smenospongia, although all the data are not presented in the paper but as additional figures.
In my opinion, Introduction should be improved. Authors could add more information about the utility of chemicals isolated from Sponge and other marine organisms to kill cancer cells, or even other therapeutic qualities, future is in the oceans.
Minor corrections:
INTRODUCTION:
1.- Line 43 :
Is there any reference about Smenamide and Calu-1 cells from 2013?
RESULTS:
2.- Lines 67-73:
I think that this paragraph suits better at Material and Methods.
3.- Antriproliferative activity
Graphics from MCF7 and MDAMB231 are clear but authors should show data on cell growth in a non tumour cell line in order to compare the effect with the two breast cancer cell lines.
Moreover, data from Figure S16 should be in the text, and no as a supplementary figure.
REFERENCES:
4.- In my opinion, there are too many references from the authors. They cite them themselves at least 11 times.
Author Response
Reviewer 2
- Line 43 : Is there any reference about Smenamide and Calu-1 cells from 2013?Line 43: the citation has been added.
- Lines 67-73: I think that this paragraph suits better at Material and Methods.Lines 67-73: in this paragraph, authors wish to give a general overview of the isolation procedure that was described in details in "Material and Methods" (line 242-260), following Marine Drugs instructions for authors guide lines.
Graphics from MCF7 and MDAMB231 are clear but authors should show data on cell growth in a non tumour cell line in order to compare the effect with the two breast cancer cell lines.Moreover, data from Figure S16 should be in the text, and no as a supplementary figure.Figure S16 has been moved from the SI to the main text.
- In my opinion, there are too many references from the authors. They cite them themselves at least 11 times.Authors have been worked on this sponge species since 2013 and got 8 papers until now. Moreover, they have been worked on Caribbean marine sponges and bioactive secondary metabolites since 1993.
Reviewer 3 Report
This is an interesting, well-written and very carefully prepared paper, which deserves for publication in the Marine Drugs. The Authors, applying molecular networking in the dereplication strategy, identified two new compounds of marine origin belonging to the class of smenamides: smenamide F and G. Despite very small amounts of the material, they were able to perform a full structural analysis of the new compounds (2D techniques of NMR, MS, ECD). The newly resolved structures were compared to already known structures of smenamides A-E. The results are presented in a convincing way, however, there were some doubts about absolute configuration at C8 and its correlation with the fragment C13/C15/C/16. The Authors assumed that configuration of smenamide F is per analogiam the same as that of smenamide A. I agree, this seems to be a rather correct assumption, but one must remember this is just a tentative ascription.
The newly discovered natural products, smenamide F and G, were evaluated for their antiproliferative properties in three cancer cell (MCF-7, MDA-MB-231 and MG-630. Both compounds demonstrated some moderate activity in two cell lines. Understanding, that this was a very preliminary screening, to improve quality of the Author’s papers in the future, I would like to consider the following point. Potential drugs should not only have high inhibitory (e.g. antimicrobial, anticancer, etc.) activity (MIC50 values), but also should exhibit high selectivity in their action. Therefore, they must be tested for their cytotoxicity in healthy human cell lines as well (MCC50 values), e.g. non-cancerous lung fibroblast line (MRC-5). The comparison of results obtained (MCC50/MIC50) allows for calculation of selectivity indices (SI) – and these values are meaningful. So, the results of biological assay are rather worthless without the reference.
My final conclusion is: ACCEPT
Author Response
Reviewer 3
Thanks for the careful consideration of our work.
We wish to underline that the configuration at C-8 was determined by Marfey’s method (line 145).
Moreover, authors understand that to go further from a preliminary screening to more interesting biological results, they need to perform cell growth experiments on a non tumour cell lines. They have planned to re-isolate all metabolites elucidated until now and study their biological activities as soon as a new collection will be possible.